# Increased performance of DNA metabarcoding of macroinvertebrates by taxonomic sorting

Kevin K. Beentjes[1,2]*, Arjen G. C. L. Speksnijder[1], Menno Schilthuizen[1,2], Marten Hoogeveen[1], Rob Pastoor[1], Berry B. van der Hoorn[1]

1 Naturalis Biodiversity Center, Leiden, the Netherlands, 2 Institute of Biology Leiden, Leiden University, Leiden, the Netherlands

* kevin.beentjes@naturalis.nl

## Abstract

DNA-based identification through the use of metabarcoding has been proposed as the next step in the monitoring of biological communities, such as those assessed under the Water Framework Directive (WFD). Advances have been made in the field of metabarcoding, but challenges remain when using complex samples. Uneven biomass distributions, preferential amplification and reference database deficiencies can all lead to discrepancies between morphological and DNA-based taxa lists. The effects of different taxonomic groups on these issues remain understudied. By metabarcoding WFD monitoring samples, we analyzed six different taxonomic groups of freshwater organisms, both separately and combined. Identifications based on metabarcoding data were compared directly to morphological assessments performed under the WFD. The diversity of taxa for both morphological and DNA-based assessments was similar, although large differences were observed in some samples. The overlap between the two taxon lists was 56.8% on average across all taxa, and was highest for Crustacea, Heteroptera, and Coleoptera, and lowest for Annelida and Mollusca. Taxonomic sorting in six basic groups before DNA extraction and amplification improved taxon recovery by 46.5%. The impact on ecological quality ratio (EQR) scoring was considerable when replacing morphology with DNA-based identifications, but there was a high correlation when only replacing a single taxonomic group with molecular data. Different taxonomic groups provide their own challenges and benefits. Some groups might benefit from a more consistent and robust method of identification. Others present difficulties in molecular processing, due to uneven biomass distributions, large genetic diversity or shortcomings of the reference database. Sorting samples into basic taxonomic groups that require little taxonomic knowledge greatly improves the recovery of taxa with metabarcoding. Current standards for EQR monitoring may not be easily replaced completely with molecular strategies, but the effectiveness of molecular methods opens up the way for a paradigm shift in biomonitoring.

**Data Availability Statement:** All relevant data are within the paper and its Supporting Information files. Raw sequence data is available from the NCBI

Sequence Read Archive (Bioproject accession PRJNA550542).

**Funding:** This study was part of the DNA Waterscan project, funded by the Gieskes-Strijbis Fonds (https://gieskesstrijbisfonds.nl/). The funder provided support in the form of material costs and the salaries for authors KKB, MH and RP. The funders had no role in study design, data collection and analysis, decision to publish, or preparation of the manuscript.

**Competing interests:** The authors have declared that no competing interests exist.

## Introduction

Now that the use of DNA barcoding for the identification of species [1] has proven its merit, research is shifting towards the integration of molecular identifications in ecological and biodiversity assessments across different biomes [2–5]. Integration of molecular techniques can provide a significant added value for the monitoring of biological quality elements (BQEs) in fields such as the quality monitoring of freshwater under the European Framework Directive (WFD) [6]. To date, many of the BQEs analyzed for WFD monitoring are still assessed using traditional morphology-based methods [7]. These traditional methods, however, are known to be hampered by difficulties in identification and substantial differences between assessors [8–10], and can be expensive due to their time-consuming nature [11–13].

Recent advances have shown the efficacy of DNA metabarcoding to assess macroinvertebrate samples [5,14] and to obtain metrics for bioassessments [15–17]. Although DNA-based methods are generally perceived as an improvement over the traditional morphological assessments [18], challenges remain to be solved before DNA-based methods can be fully incorporated into routine bio-monitoring. Studies employing metabarcoding of aquatic macroinvertebrates are often limited to single samples [19], a select subset of taxa [20] or rely on mock communities [21–24]. Research that does cover a broader variety of WFD monitoring samples often deals with differences in taxonomic resolution between morphological and DNA analyses [16,25]. One of the main confounding effects in the use of molecular approaches is the effect of primer bias and preferential amplification in complex samples, leading to taxonomic bias [26,27]. Interactions between taxa from various organism groups, of varying sizes and in varying biomass ratios remain understudied, and implications can be severe, limiting the possibility to relate metabarcoding read data to actual taxon abundances [28], even though these actual abundances might not be as important for simple ecological quality ratio calculations used by water monitoring agencies [29].

In this paper, we assess the implementation of DNA metabarcoding for species identification in bulk samples collected under the WFD. We evaluate the performance of DNA metabarcoding-based identification of taxa across six different taxonomic groups that collectively cover most of the traditional macroinvertebrate samples collected for WFD freshwater quality assessments: Annelida, Crustacea, Heteroptera/Coleoptera, Mollusca, Trichoptera/Odonata/Ephemeroptera, and Diptera. Our aim is to assess the effects of taxonomic sorting on the recovery of taxa from bulk metabarcoding, and the impact of replacing these groups with molecular data on ecological quality ratio (EQR) scoring. While EQRs are a simplified way to look at community compositions, they provide an insight in water quality, and are widely used by water monitoring agencies to assess the status of surface waters under the WFD [7,29]. We also discuss some concerns on DNA reference databases that may hinder successful application of molecular methodology in biomonitoring.

## Materials and methods

### Sample selection and processing

Freshwater macroinvertebrate samples were collected in the Hoogheemraadschap Rijnland monitoring district in 2010 and 2012 by ecological survey company Aquon (Leiden, the Netherlands). Samples were collected and analyzed according to standardized WFD monitoring guidelines [30]. Specimens were sorted by Aquon taxonomists into seven different categories during morphological analysis, and stored separately in ethanol per taxon group: ANNE (Annelida), ACA (Hydrachnidia, stored in Koenike's fluid), CRUS (Crustacea), HECO (Heteroptera and Coleoptera), MOLL (Mollusca), TOE (Trichoptera, Odonata and

Ephemeroptera), and REST (miscellaneous, predominantly Chironomidae and other Diptera). Specimens were identified to lowest possible level, preferably species level. For this study, we selected 25 samples out of 138 from the monitoring cycles of 2010 and 2012. More recent samples could not be used, as there is a five-year retention period for WFD monitoring samples. Samples were selected based on the WFD ecological quality ratio (EQR) scores (range 0.158–0.759), as well as the Shannon-index (range 0.840–4.326), to represent a broad range of sample diversities and complexities (for all 138 samples, EQR ranged from 0.059 to 0.847 and Shannon-index ranged from 0.602 to 4.326). EQR scores in the Dutch WFD monitoring range from 0.0 to 1.0, and are divided into 5 categories ranging from "bad" (EQR 0.0–0.2) to "high"(-EQR 0.8–1.0) (for more detail, see [25]). The 25 selected samples represented four out of five quality classes, in the 138 samples there was only one sample that was scored as "high". The full taxon lists with specimen counts have been included in the supplementary data (S1 File).

Not all of the seven groups were present in all samples. The water mites (ACA) were excluded from the analysis, as they were preserved in Koenike's fluid (45% water, 45%, glycerin, 10% glacial acid acetic), which had a negative impact on the preservation of DNA and we were unable to obtain useable DNA extracts from the samples. To account for the missing taxa, water mites were also removed from the morphological lists during the comparison of DNA and morphology.

## DNA extraction and amplification

Specimens were homogenized in 15 ml sterile tubes containing 10 steel beads (5 mm diameter), using the IKA Ultra Turrax Tube Drive (IKA, Staufen, Germany) in a fixed volume of 5.0 ml 96% ethanol. Each tube was ground three times for one minute on the maximum speed setting (6000 rpm). A tube with only 5.0 ml of 96% ethanol was used as an extraction blank. After homogenization, 500 μl of the ethanol with ground specimens was transferred to a 2 ml tube, and the ethanol was evaporated using a Concentrator plus vacuum centrifuge (Eppendorf, Nijmegen, the Netherlands). DNA was extracted from the remaining dry debris using the Nucleomag 96 Tissue kit (Macherey-Nagel, Düren, Germany) on the Kingfisher Flex Purification System (Thermo Fisher, Waltham, MA, US), with a final elution in 150 μl. To simulate a total DNA extraction on all taxa of one sampling location combined, 5.0 μl of DNA extract from each of the taxonomically sorted samples belonging to one location was combined into a pool, which was amplified and sequences in the same way as the sorted samples.

A two-step PCR protocol was used to create a dual index amplicon library, using primers BF1 and BR2 [22] to amplify a 316 base pair fragment of the COI barcoding region. These primers have been shown to successfully amplify a wide range of freshwater macroinvertebrates. All 183 samples (158 individually extracted tubes and 25 pools) were amplified and labeled separately, using two PCR replicates for each sample. First round PCRs were performed in 20 μl reactions containing 1x Phire Green Reaction Buffer, 10 μg BSA (Promega, Madison, WI, US), 0.5 mM dNTPs, 0.4 μl Phire Hot Start II DNA Polymerase (Thermo Fisher, Waltham, MA, US), 0.65 μM of each primer and 2.0 μl of template DNA. Initial denaturation was performed at 98°C for 30 seconds, followed by 30 cycles at 98°C for 5 seconds, 50°C for 5 seconds and 72°C for 15 seconds, followed by final elongation at 72°C for 5 minutes. PCR success was checked on an E-Gel 96 pre-cast agarose gel (Thermo Fisher, Waltham, MA, USA). PCR products were then cleaned with a one-sided size selection using NucleoMag NGS-Beads (Macherey-Nagel, Düren, Germany), at a 1:0.9 ratio.

Second round PCRs to add the individual P5 and P7 Illumina labels (Nextera XT Index Kit; Illumina, San Diego, CA, USA) were performed using 3.0 μl of cleaned PCR product from the first round in a 20 μl reaction containing 1x TaqMan Environmental Master Mix 2.0 (Thermo

Fisher, Waltham, MA, USA) and 0.5 μM of each primer. Initial denaturation was performed at 95˚C for 10 minutes, followed by 14 cycles at 95˚C for 30 seconds, 55˚C for 60 seconds and 72˚C for 30 seconds, followed by final elongation at 72˚C for 7 minutes. All PCRs were performed in 96-well plates, with replicates in separate plates. Each plate contained two wells with an artificial internal control (AIC) sample that was used to gauge the amount of cross-contamination between samples in the amplification process in the laboratory. The artificial control was based on the COI barcode region of a Reeve's muntjac (*Muntiacus reevesi*) with several primer sets built into the sequence, and synthesized by IDT (Leuven, Belgium) (S1 Fig). Second round PCR products were quantified on the QIAxcel (Qiagen, Venlo, the Netherlands) and pooled equimolarly per PCR plate using the QIAgility (Qiagen, Venlo, the Netherlands). Pools were cleaned with a one-sided size selection using NucleoMag NGS-Beads (ratio 1:0.9) then quantified on the Bioanalyzer 2100 (Agilent Technologies, Santa Clara, CA, USA) with the DNA High Sensitivity Kit. The pools were then combined equimolarly into one sample and sequenced in one run of Illumina MiSeq (v3 Kit, 2x300 paired-end) at Baseclear (Leiden, the Netherlands). Sequence data is available from the NCBI Sequence Read Archive (Bioproject accession PRJNA550542).

## Bioinformatics

Quality filtering and clustering of the entire dataset was performed in a custom pipeline on the OpenStack environment of Naturalis Biodiversity Center through a Galaxy instance [31]. Raw sequences were merged using FLASH v1.2.11 [32] (minimum overlap 50, mismatch ratio 0.2); non-merged reads were discarded. Primers were trimmed from both ends of the merged reads using Cutadapt v1.16 [33] (minimum match 10, mismatch ratio 0.2). Any read without both primers present and anchored was discarded. PRINSEQ v0.20.4 [34] was used to remove reads with length below 313 bp and above 319 bp, to allow for natural variations in coding sequence as well as potential primer slippage [35]. Sequences were dereplicated and clustered into Molecular Operational Taxonomic Units (MOTUs) using VSEARCH v2.10.3 [36] with a cluster identity of 98% and a minimal accepted abundance of 2 before clustering. The presence of AIC reads in the regular (non-control) samples, as well as the presence of non-AIC DNA in the control samples was used to determine the MOTU filtering threshold; only MOTUs with read abundances above 0.025% were retained for each replicate. Samples with fewer than 4,000 reads were discarded and PCR replicates were combined according to the additive strategy, counting all MOTUs, irrespective of how many replicates they occurred in [37], as the intent was to recover as many taxa as possible.

MOTU sequences were compared to a custom reference database using an extended BLAST+ script (https://github.com/naturalis/galaxy-tool-BLAST). The custom reference dataset included 2,757 COI barcodes obtained from WFD species collected in the Netherlands as part of the national DNA barcoding campaign [38], supplemented with sequences obtained from BOLD [39] belonging to the 795 genera listed on the Dutch WFD species list. A total of 350,449 public sequences of 679 genera were retrieved from BOLD using the package 'bold' [40] in R [41] (sequences downloaded 28 June 2018). The remaining genera were either not present in the BOLD database (107 genera) or had no public sequences linked to them (9 genera). The exclusion of sequences not identified to at least genus level allowed for linking taxa to the Dutch Species Register (https://www.nederlandsesoorten.nl/) based on genus names, making all taxonomic data compatible for use in lowest common ancestor analysis. The final database was dereplicated, removing all entries that had 100% identical DNA sequences and species names. MOTUs were also compared to a second custom reference library containing COI sequences and bacterial genomes downloaded from NCBI GenBank [42] (sequences

downloaded 21 August 2018), to help filter out non-macroinvertebrate MOTUs and correct for misidentifications based on contaminated (e.g. *Homo sapiens* or *Wolbachia*) or otherwise erroneous sequences in the BOLD database.

The top 100 hits were obtained for both BLAST comparisons. Anticipating gaps in the DNA database, we developed a custom lowest common ancestor (LCA) tool to be able to assign higher-level taxonomic assignments for MOTUs without direct hits (>98% match and 100% coverage) in the reference database. The LCA tool was based on MEGAN [43], with adaptations to allow for the use of custom taxonomic databases and integration into the Galaxy infrastructure (https://github.com/naturalis/galaxy-tool-lca). The LCA script was performed on the top 5% hits, with bit-score >170, a minimum identity of 80% and a minimum coverage of 80%. The LCA tool was set to identify MOTUs no further than genus level. All direct hits (>98% match) were retrieved directly and accumulated based on taxon name associated with the sequences. To check for non-Dutch taxa and synonyms, a custom taxon matcher tool (https://github.com/naturalis/galaxy-tool-taxonmatcher) was used to compare all the names obtained to taxa recorded in the Dutch Species Register. In case of multiple taxa having a direct hit, the names were manually checked and taxonomy was determined based on the following set of rules: (1) non-Dutch species were removed, (2) synonyms were resolved, (3) sub-species level identifications were set to species level, (4) when a MOTU matched both genus level sequences and species level reference sequences of the same genus, species level identifications were retained, (5) putative misidentifications or contaminations were removed, based on expert judgment and the top 100 BLAST hits, (6) if one species matched consistently higher than another, the species with a better match was retained, (7) in case of equal matches with multiple species, all species names were retained (e.g. species complexes that could not be resolved with the available reference sequences).

## Comparison morphology versus molecular identification

After applying the LCA script, MOTUs with the same taxonomic assignment were aggregated. Individual samples were then accumulated into their respective locations, with exception of the pool sample. Taxa lists obtained from the molecular analysis were compared to the WFD taxa lists based on conventional morphological identifications provided by Aquon. Morphological taxa lists were first matched to the Dutch Species Register using the same script that was used to compare the taxa lists retrieved from metabarcoding, to make the species names in both lists compatible. Before the comparison, redundancy was removed from both taxa lists, to exclude uncertainties in identifications or potential duplicates (i.e., a genus level identification was omitted if the list also contained specimens from that genus that were identified to species level).

DNA-based taxon lists from the pools and the separately sequenced samples added together were both compared to the morphological list manually. Each entry on the combined lists was classed into one of the following categories: (1) "found", where there was an exact match between both lists; (2) "identified at a different level", when there was a match, but either one of the lists had a higher-level identification; (3), "putative misidentification", in cases where two different species from the same genus were listed on the respective lists; (4) "missing in reference" when the morphologically identified species was not covered by the DNA reference database; (5) "not found", when the taxon was covered in the reference database, but only encountered in the morphological list; (6) "extra", when the taxon was only encountered in the DNA list. To calculate the overlap between morphology and DNA, the first three categories were grouped together as being found in both lists, the taxa missing from the reference were counted towards the taxa only found in the morphology.

To analyze if uneven sequencing depth between samples pooled prior to amplification and the separately sequenced samples added together had any effect on taxonomic recovery, and to allow for better comparison between samples, all data was rarefied to the lowest read count available. Pooled samples were all rarefied to 15,000 reads, separately sequence samples representing the different taxon groups were each rarefied to 2,500 reads to adjust for the fact that most pools consisted of six taxon groups.

Ecological quality ratio (EQR) scores were calculated according to the Dutch standards for both morphological and DNA-based taxon lists, using the QBWat software version 5.33 [44] (with redundancies removed as described previously). Scores were calculated based on presence/absence data (with all specimen counts set to one) for both morphological and molecular data. Previous research has shown that abundances had limited impact on the EQR score [29].

## Results

### Sequence run statistics

Sequencing resulted in a total of 9,998,809 read pairs. After merging and quality filtering, 9,081,986 sequences were retained for MOTU clustering. AIC reads were detected in several non-control samples. A 0.025% threshold for filtering low-abundance MOTUs from each sample removed control reads from all samples. After filtering the MOTU table 2,460 MOTUs were retained in the non-control samples, representing 8,200,488 reads. Out of 366 replicates (158 sorted samples, 25 pools, all in duplicate), 77 with fewer than 4,000 reads were discarded. On average, PCR replicates had 28,345 reads (range 4,197–69,919), and 43.0 MOTUs (range 2–132). There was no correlation between number of reads and number of MOTUs in each sample.

### Taxonomic composition

Using the two reference libraries, 1,837 MOTUs were identified as macrofauna taxa listed on the Dutch WFD taxon list on at least order level. A total of 319 MOTUs had direct matches above 98% percent, representing 213 distinct species or species complexes. The remaining MOTUs were identified to genus (1,394 MOTUs, 121 genera), family (93 MOTUs, 12 families) or order level (31 MOTUs, 11 orders). MOTUs that were not identified to at least order level were discarded. The final dataset of the sorted and separately amplified groups represented 208 species, 159 genera, 75 families and 34 orders. The data for the pools that were combined before the PCR amplification contained 172 species, 139 genera, 65 families, and 31 orders. The morphological lists covered 214 species, 151 genera, 73 families, and 30 orders (excluding the water mites) (S1 File). DNA-based taxon richness was significantly correlated with morphological taxon richness for both sorted samples (r = 0.662, p = 0.001) and pooled samples (r = 0.602, p = 0.002), where redundant taxa had been removed (Fig 1A). An additional 13 macroinvertebrates identified at species level were lost by the 0.025% threshold filtering (and only observed in the data that was discarded by this filter step). Seven of these were also recorded in the morphological assessment, the other six were only found using DNA. One of the species found in the discarded DNA-based data was *Musculium lacustre*, which was present in four samples where it was also detected morphologically, but only with one or two reads in each case.

To exclude the influence of sequencing depth (as sorted samples combined represented more sequencing depth than the pooled samples), we rarefied the samples to such an extent that sorted samples represented only one sixth of the pooled samples (most pools consisted of six combined extracts). Without rarefaction, the sorted samples had an average of 272,914 reads (range 170,637–424,726), which was 4.8 times more than the pools had (57,283 on

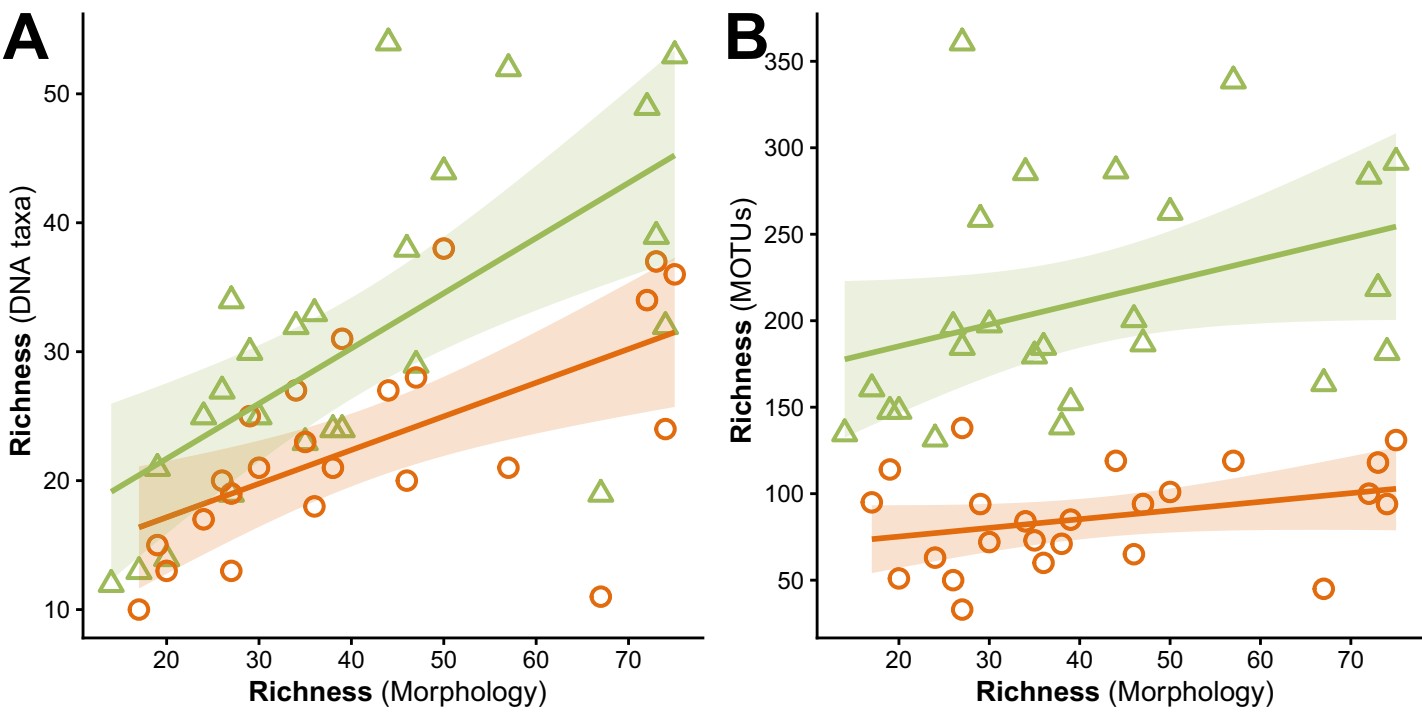

**Fig 1. Taxon and MOTU richness.** Relation between the morphological richness of samples and the (A) DNA taxon richness and (B) MOTU richness, for both the sorted samples (green triangles) and the pooled samples (orange circles), with a 95% confidence interval. Taxon richness was based on the taxon lists where redundant taxa had been removed. Correlations were significant for the taxon richness for both sorted samples (r = 0.662, p = 0.001) and pooled samples (r = 0.602, p = 0.002), but not for MOTU richness (r = 0.365, p = 0.072 and r = 0.331, p = 0.115, respectively).

average, range 15,061–122,002). They also had 2.67 times as many MOTUs and 1.52 times as many taxa as the pools. With rarefaction the sorted samples still had 2.22 times as many MOTUs and 1.40 times as many taxa; neither was significantly lower than without rarefaction.

### Comparison morphology versus molecular identification

Retaining the redundant taxa, the average richness of pooled samples (32.5 on average, range 16–56) was significantly lower than that of the sorted samples (47.6 on average, range 22–76) (Dunn's test, p = 0.005). When redundant taxa were removed, the richness of the pooled samples (22.9 on average, range 10–38) was again lower than the sorted samples (30.6 on average, range 12–54), but not significantly. Compared to the morphological richness with redundant taxa (46.7 on average, range 16–89), the richness of the pooled samples was significantly lower (Dunn's test, p = 0.027). The richness of the pooled samples was also significantly lower than the morphological richness when redundancy was removed (40.8 on average, range 14–75) (Dunn's test, p < 0.001). The richness of the sorted samples was not significantly different from the morphological richness in either situation.

For 13 out of 24 separately processed mollusc samples (one sample did not include molluscs) we were unable to amplify molluscs using the standard approach for DNA extraction and PCR. Additionally, four annelid samples, three Heteroptera/Coleoptera (HECO) samples, one crustacean sample, and one TOE sample failed to amplify, although the latter two only contained three and two species, respectively. The failed mollusc samples on average contained 13.2 morphologically identified taxa (range 5–20), the failed HECO samples 18.3 taxa (range 7–25) and the missing annelids accounted for 6.3 taxa (range 2–12). If taxa from the failed

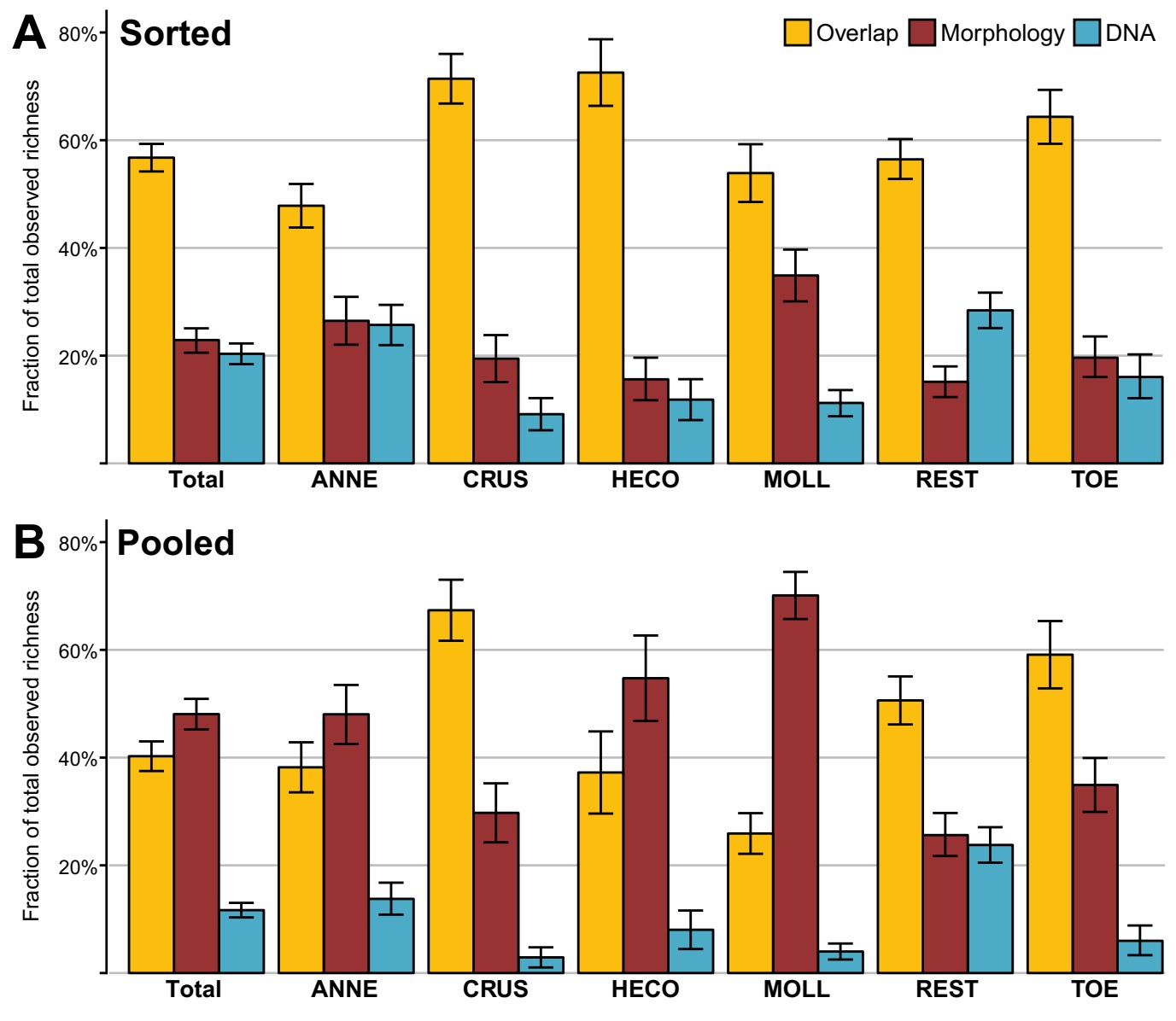

**Fig 2. Overlap between morphological and DNA-based taxa.** The fractions of total observed diversity present in both morphological assessment and DNA-based methods (yellow) and the fractions only represented in morphology (red) and DNA (blue), for the sorted samples (A) and pooled DNA analysis (B). The fractions were also assessed for each of the sorted taxa groups separately according to the following sorting of taxa: ANNE (annelids), CRUS (crustaceans), HECO (Heteroptera and Coleoptera), MOLL (molluscs), REST (rest groups, almost exclusively chironomids and other dipterans) and TOE (Trichoptera, Odonata and, Ephemeroptera). Error bars indicate the standard error.

samples are excluded from the analysis (as they can only count towards the fraction of taxa not found by DNA), the overlap between the taxon list from sorted samples added together and the morphological taxon list was 56.8% on average (range 32.5–91.7%). On average, 22.9% of taxa were only found in morphology (range 0–50.0%), and 20.3% were only recovered using DNA (range 5.6–35.0%) (Fig 2A). If failed samples are included, the overlap between morphology and DNA was 47.6% on average (range 22.9–73.3%). For the pooled samples, the combined taxon lists contained an average of 47.3 taxa, with a 40.3% overlap between morphology

and DNA (range 13.0–62.8%). 48.1% of taxa were only recorded in the morphological list (range 27.9–84.1%), and only 11.7% were found exclusively with DNA (range 0.0–27.8%) (Fig 2B). In 14 out of 24 samples (one pooled sample failed to amplify), the fraction of taxa only found with morphology was higher than the fraction of overlap between the two taxon lists, the fraction of taxa only found using DNA was never higher than the fraction of taxa found only in the morphological analysis. In contrast, in 14 out of 25 samples where taxa were sorted prior to DNA analysis, the fraction of taxa exclusively found with DNA was higher than the morphology-only fraction.

The three categories that were counted towards the overlap contained 402 entries (72.6%) where there was a direct match between the species recorded in the morphological analysis, and the species identification obtained from metabarcoding. In 124 cases (22.4%) there was a match between morphology and metabarcoding, but the entries on both lists were not identified to the same taxonomic level. The majority of these were annelids not covered in the reference database at species level (but were identified from molecular data at higher level using LCA) and dipterans identified to species level in the metabarcoding analysis but only identified at genus level or higher in the morphological data. The remainder were 28 cases of putative misidentifications (5.1%), where both list contained a different species from the same genus.

Looking at the six taxa groups separately (again excluding the failed samples), the overlap varies. The highest overlap was found in the crustaceans and HECO samples (71.4% and 72.6%, respectively), even though for HECO in one case the morphological and DNA-based taxon lists did not overlap at all (both, however, only contained one species each). The lowest overlap was found in the annelid samples (47.8% on average). Overlap for the MOLL, REST and TOE samples was 53.9%, 56.4% and 64.3% on average, respectively (Fig 2A). For the REST samples, the fraction of taxa found only in the DNA was larger than the fraction of taxa only recorded morphologically, for all other groups there were more taxa in the morphology list than there were on the DNA list. In 18 samples, more taxa were found with DNA than with morphology, in 26 samples more taxa were obtained with morphology. For 18 samples the morphology and DNA taxon lists was a complete match, although some taxa were not identified up to the same taxonomic level for both methods. In addition to the previously mentioned HECO sample, there was one other sample in the TOE set where DNA and morphology were mutually exclusive (S2 Fig). In the pooled samples, the overlap between morphology and DNA was considerably lower for most taxa groups, but most noticeable in the HECO and mollusc samples, where most taxa were only present on the morphological list. For all groups, more taxa were found with morphology than were found with DNA metabarcoding (Fig 2B).

## Ecological quality ratios

The EQR scores based on the DNA data differed considerably from the morphology-based EQR scores for both the pooled and the sorted samples (Fig 3A and 3B). There was only a moderate correlation between the morphology- and DNA-based scores (Pearson correlation, r = 0.596 and 0.545, respectively). The scores obtained from the pooled samples were usually lower than the morphological scores (16 out of 24). For the sorted samples, half the samples (13 of 25) had a lower score using molecular identifications, the other half (12 of 25) scored higher based on DNA data. The average absolute difference in EQR score was similar for both datasets: 0.12 for the pooled samples (range 0.007–0.302) and 0.11 for the sorted samples (range 0.007–0.310). Using the pooled samples, 15 out of 24 locations scored in a different quality class (five higher, ten lower), and for the sorted samples, 12 of 25 ended up in a different quality class (five higher, seven lower). When replacing just one of the groups with molecular data for the EQR calculations, the correlations between the two scores were much stronger

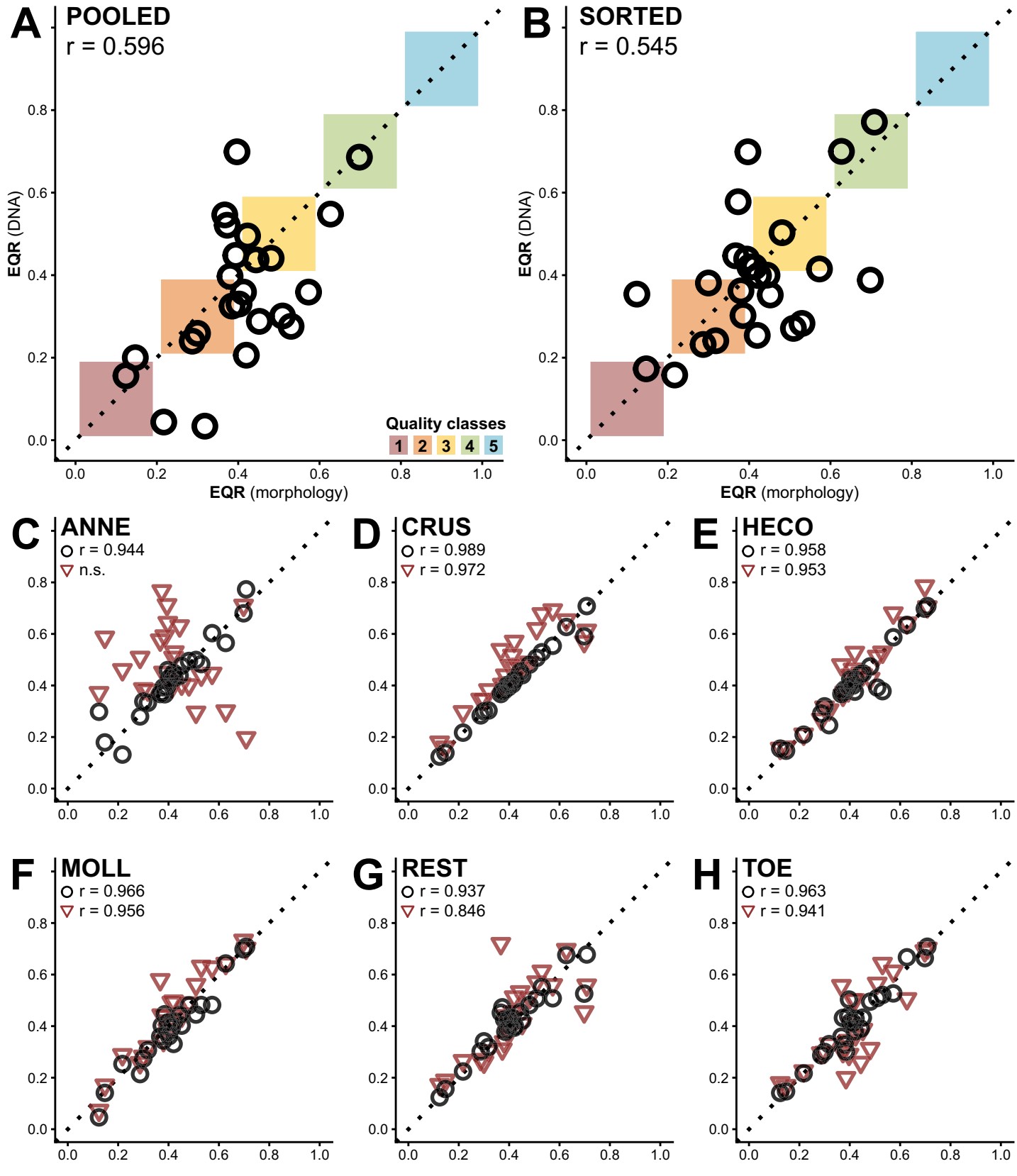

**Fig 3. EQR scores.** Comparison for the EQR score calculated on morphological data versus the score calculated on DNA data for (A) the pooled and (B) the sorted and separately sequenced samples. Both showed a moderate correlation between the scores (Pearson correlation, p = 0.002 and 0.005, respectively). Better correlations were found when only replacing one of six taxon groups with molecular data (on the y-axis): (C) Annelida, (D) Crustacea, (E) Heteroptera and Coleoptera, (F) Mollusca, (G) Chironomidae and other Diptera and (H) Trichoptera, Odonata, and Ephemeroptera (black circles, Pearson correlation values provided in the panels, p < 0.001 for all groups). To assess the influence of each of the respective groups on the EQR score, the original scores (x-axis) were also compared to EQR scores where one taxon group was completely removed (y-axis) from the taxon list before the analysis (C-H, red triangles, Pearson correlation values provided in the panels, p < 0.001 for all groups).

(ranging from r = 0.937 for REST to 0.989 for CRUS, p < 0.001 for all groups), even with the complete removal of some groups due to failed samples (Fig 3C–3H).

## Discussion

We found that pre-sorting of samples into six basic taxon groups vastly improved the recovery of taxa using metabarcoding of bulk samples, with 46.5% more taxa found as compared to the samples where DNA was pooled prior to amplification and sequencing (47.6 versus 32.5 on average). The average overlap between the morphological and molecular (for the sorted samples) taxon lists was 56.8%, with the fractions of taxa found in only the morphology and only the DNA roughly equal (22.9% and 20.3%, respectively) (Fig 2A). Discrepancies between morphology and DNA-based species lists were expected, based on missing taxa from the reference database, known difficulties with morphological identification of taxa [9,10], and primer biases [28] as contributing factors. Even though they were tested mainly on insects, the primers used in this study showed good *in silico* potential for all taxonomic groups included in our samples [22], especially compared to some other oft-used primers. While there may be primers that perform better for a specific group, a single, broad-range primer set that perform equally well on all taxa will most likely never exist [27]. Large differences were already observed between two morphological assessments in freshwater monitoring samples in previous studies, where there was more than 30% difference in identification of taxa. All taxon groups seemed to be equally prone to errors in morphological identification, even those deemed difficult to identify [45]. In our data we see that the overlap between morphology and DNA varies between the different groups, being highest for the Crustacea and the Heteroptera /Coleoptera. The poor performance of the mollusc samples may not be entirely attributable to the primers, as molluscs are the group that is most affected by differences in biomass between the different species in a sample.

There have been few studies comparing morphological identifications and DNA-based identifications on actual samples, instead of relying on mock communities. A study assessing the taxa detected by morphology and DNA on Finnish WFD samples (using the same primers as this study) found considerably more taxa with DNA than they did with morphology [16], but morphological assessments did not include species or genus level identifications for certain groups, such as the species-rich Chironomidae. The taxonomic resolution in the present study was comparable between morphology and DNA metabarcoding, and explains why richness estimations were more comparable on average. Still, we found some differences between taxon lists caused by disparity in resolution for certain taxa. On the side of the morphology, higher-level taxonomic identifications have been made due to the difficulty of distinguishing taxa, especially those in larval stages. For example, none of the Ceratopogonidae had been identified beyond family level using morphology, but five different genera were detected with DNA. On the other hand, the DNA reference database did not cover all the taxa that were listed in the morphological dataset (6.5% of the morphologically identified species had no DNA reference). For instance, every specimen of *Alboglossiphonia* was only identified up to genus level using the LCA tool in the DNA analysis, as all three species recorded in the morphological analysis

were unaccounted for in the reference database (sequences could still be identified to genus level based on matches to congeneric species).

Some groups that were examined in this study consists of considerably more taxa than others. This difference in group size inevitably leads to a larger number of "lost taxa" when one taxon dominates the reads due to the effects of preferential amplification. In the majority of the pooled samples (15 of 24) more than half of the reads is provided by one of the six groups (S3 Fig), and in eleven samples more than half the reads even belonged to a single taxon. While some have argued that for general patterns in biodiversity, the effects of primer bias may be limited, the taxonomic bias caused by primer mismatches in certain taxonomic groups can be an issue when trying to reconstruct taxa lists [27]. Taxonomic sorting can improve the recovery of taxa, as witnessed by the improved performance of the sorted and separately sequenced samples in comparison to the pooled samples, which represent a broader range taxa. In the sorted samples, 46.5% more taxa were found than in the pooled samples (47.6 versus 32.5 on average), also leading to more overlap with the morphological list (56.8% versus 40.3% on average). Similar improvements have been found when using a size-based sorting of specimens prior to DNA extraction and amplification, where around 30% more taxa were found compared to non-sorted samples [23], although others report that amplification bias across size ranges may be limited with deep sequencing [27]. When assessing the separate groups, the effect of the pooling of samples prior to DNA amplification and sequencing has the largest effect on the HECO and mollusc samples, where 65.6% and 46.6% fewer taxa were found in comparison to the sorted and separately sequenced samples (Fig 2B). Rarefaction showed that the reduced sequencing depth of the pools, when compared to the combined separately sequenced samples, was not solely responsible for the reduction in detected taxa. Even when rarefied to the same sequencing depth, we still obtained 40.0% more taxa in the sorted samples. A study assessing the taxonomic recovery of tropical forest arthropod communities showed similar findings, where there was some decline in MOTUs recovered for specific taxon groups in increasingly complex mixtures. This was mostly caused by the introduction of other groups, which were apparently amplified preferentially [27], comparable to our observations with HECO and mollusc taxa. Taxonomic sorting into the groups presented in this study is relatively straightforward and would require only superficial knowledge of taxonomy. Compared to genus or species level sorting and identification, both the time and costs involved are between one and two orders of magnitude lower [13,46].

The difficulties in identifying specimens using morphology can also express themselves in the DNA-based identities, by way of having erroneously identified specimens within the DNA reference library. We encountered a variety of unresolved taxa and putative identification errors in the reference data downloaded from BOLD. In the 350,449 public sequences we found 554 cases where congeneric species had identical sequences. These are not necessarily identification errors, as some closely-related species are known to be indistinguishable by the DNA barcode region [47], but do highlight the need to not look at just the "top 1" or "best hit" matches when comparing sequences to a reference database. When multiple hits with the same scores are found, matching algorithms do not always consistently place the same match at the top of the list, introducing random variation between analyses when only looking at the first hit. Additionally, 47 cases of identical sequences with different species from different genera were found, some of which could be traced back to actual contaminated sequences (e.g. *Homo sapiens* or *Wolbachia*). Most of such misidentified records have been flagged by BOLD curators, which was verified by manually checking a random selection of records. Moreover, recent analysis of the BOLD data revealed a relatively high number of specimens that had been identified using "reverse BIN taxonomy", adding further levels of uncertainty to the reference datasets retrieved from BOLD [48]. To improve the use of public data such as the sequences

deposited in BOLD, the ability to filter data based on record flags or identification method is essential.

An incomplete reference databases is a major issue that limits the use of metabarcoding for species identification [49,50]. While 93.5% of the morphologically identified species of this study had reference sequences, database coverage for all Dutch WFD taxa is only 86.1%. There are large differences for each of the taxa groups as defined by this study, with 63.5% of annelids covered by reference sequences (54 of 85 species), while 96.5% of the TOE group has been barcoded (273 out of 283 species). Additionally, we still observe difficulties in identification for species known to have high genetic diversity. For example, 14 MOTUs were identified as *Asellus aquaticus*, another 109 were identified at genus level as *Asellus* (for which only *A. aquaticus* is recorded in the Netherlands). The tendency to overestimate richness based solely on MOTUs (see also Fig 1) has already been reported in the past, with population and haplotype differences increasing richness estimates [25,51]. The length threshold used in this study (allowing for sequences which were three base pairs shorter or longer than the 316 bp target to pass quality filtering) may have contributed to an overestimation of richness based on MOTUs. We aimed to mitigate this effect by aggregating all MOTUs with identical taxonomic identification and discarding any unidentified MOTU from the analysis. While alternate clustering methods may exaggerate or downplay this effect of overestimation, the difference in intraspecific variation between taxonomic groups will lead to either overestimations for taxa with high intraspecific variation or underestimations by lumping taxa with low interspecific variation depending on the cluster settings. The observed variation also suggests that the DNA reference library could be improved by better geographical coverage, incorporating a wider range of haplotype variation. Another phenomenon that may have caused an overabundance of *Asellus* and other genera in the MOTUs, is the presence of pseudogenes that have been amplified [52,53], especially with deep sequencing of highly abundant taxa. Many of the MOTUs identified at genus level have fewer reads (1,768 on average) compared to the MOTUs with species level identification (75,128 reads on average). Similar patterns were seen for other genera as well (e.g. *Helobdella*, *Limnomysis*), including genera that have more than one species recorded for the Netherlands, and which were all represented in the reference database (e.g. *Cymatia*, *Erythromma*, *Noterus*).

Haplotypes and pseudogenes aside, we should be wary of the fact that many taxon groups still contain undescribed diversity and cryptic species [54], which may be perceived as overestimations of taxa when using DNA-based identification methods. This information can still be valuable, as it has been shown in mayflies that cryptic species exhibit a wide variety of tolerances and responses to ecosystem stressors [55]. Furthermore, MOTU level analysis of Chironomidae has demonstrated that even without binomial names, different putative taxa could be identified, showing different response patterns [56]. This opens possibilities to use DNA-based delimitations for comparative quality assessments and impact studies even for those taxon groups that are poorly defined in reference databases, which is still hampering the use of DNA-based identification in various groups [57]. DNA-based identification may not always exactly reflect the observations made by traditional morphological methods, but at least may provide a more consistent way of identifying taxa [18]. Morphological assignments are prone to discrepancies between assessors, as shown by large differences between identification made in audits of WFD assessments [9,10]. The choice of tools and parameters used in the processing of raw sequence data (such as filtering and clustering) can have a significant impact on taxonomic inferences as well [37,58], but in comparison with morphological assessments should be easier to report and standardize. Molecular data is also more easily re-analyzed when new insights are developed, and is backwards compatible with updated reference databases.

The impact of DNA-based identifications on the EQR scoring is considerable, for both the pooled and the sorted samples, with neither giving a better approximation of the morphology-based score (Fig 3A and 3B). The correlation between the two scores was only moderate (Pearson correlation, r = 0.596 and 0.545, respectively). EQR scores for the pooled samples were generally lower than the morphology-based EQRs (16 out of 24), whereas the sorted samples provided scores that were lower in half the samples, higher in the other half. The average absolute difference between the EQRs obtained from morphological data and the DNA-based scores was similar for both datasets (0.12 and 0.11 for the pooled and sorted samples, respectively). When replacing just one of the six groups from the morphological taxon list with DNA-based identifications, the impact on the EQR score was considerably lower. This allows for the use of DNA metabarcoding for a select group of taxa, for example in cases where morphological assessments are difficult or time-consuming (such as Chironomidae), without the need to recalibrate the entire EQR scoring method. In such cases, it would also be possible to use primers that are tailored more specifically to the investigated taxa, in order to limit primer bias. The largest deviation was seen in the mollusc samples, where the absolute difference was 0.033 on average (range 0–0.091), but the scores were still strongly correlated (Pearson correlation, r = 0.966, p < 0.001). Molluscs were also the group for which most samples failed to amplify (13 out of 24, S2 Fig), but this did not seem to have too much of an impact on the scoring. However, complete removal of groups can have a substantial impact on the EQR score, especially for the annelids (Fig 3C–3H). Removal of the water mites, which were excluded from the analysis due to inability to obtain DNA, had a comparable impact on the EQR scoring to some of the other groups (S4 Fig). To minimize the effect of stacking these impacts, the water mites were completely discarded from all EQR analyses.

The fact that DNA-based EQR scores are so different from the scores based on traditional morphological surveys can partly be attributed to the changes in taxonomic resolution and deviations between the two taxon lists, but the differences are likely exaggerated by the changes in richness as well. For the Dutch EQR calculations, the percentage of characteristic taxa, and positive and negative indicator species (as a fraction of the total richness) play an important role for the final score [29,59]. While the average taxon richness was not significantly different between the morphological assessment and the sorted and separately sequenced samples used for the DNA-based calculations, the differences for each sample were considerable (Fig 1A), with an average difference in richness between morphology and DNA of 12.0 (range 1–48). Together, the changes in the number of negative and positive indicators, and the changes in the ratios of these indicators can have a significant impact on the final EQR score (Fig 3A and 3B).

Molecular techniques may not directly replace traditional morphology under the current WFD monitoring standards, and future monitoring requires a paradigm shift to fully incorporate the potential of DNA-based methodology. Time is needed for new techniques to prove their worth in the field of biomonitoring and be accepted by monitoring agencies and policy makers. Being able to replace morphological assessment for only one or a few taxon groups without needing to redefine the framework for BQE monitoring, opens possibilities for gradual implementation of DNA-based identifications for those groups that are most difficult to identify, time-consuming or where taxonomic expertise is getting scarce.

## Conclusions

There are considerable differences when directly comparing the outcome of traditional morphological assessment and DNA metabarcoding-based identifications of bulk samples, including their effects on the EQR score under current standards. Our data shows that DNA metabarcoding compares better to morphological assessments for some taxonomic groups

than for others, partly based on the underlying DNA reference database, or lack thereof. Mismatches were observed between morphology and metabarcoding, but the latter will be less reliant on individual biases introduced by different assessors, and therefore lead to more consistent assessments. Taxonomic sorting into basic groups improves the taxon recovery, as shown in this study, where 46.5% more taxa were found when samples were sorted into six basic groups prior to DNA amplification and sequencing. Even when corrected for sequencing depth, sorted samples still produce around 40% more taxa as non-sorted samples. DNA-based assessments may not directly replace traditional monitoring in the near future, but can certainly contribute to the current methodology, especially for those groups that are perceived as difficult to identify, to allow for more consistent and faster identifications. Metabarcoding would greatly improve with addition of vouchered specimens to reference databases. Furthermore, we show that replacing only one of six taxa groups assessed in this study by molecular data has limited impact on the EQR scoring, opening possibilities for gradual replacement of traditional identification, or supplementing the traditional identification with DNA-based tools, which will help with the acceptance of molecular methodology in WFD monitoring.

## Supporting information

**S1 Fig. Artificial internal control.** The artificial control (AIC) used to measure cross-contamination. The sequence is based on the COI barcode region of a Reeve's muntjac (*Muntiacus reevesi*) with several primer sets built into the sequence (forward strand shown). Binding sites for COI primers are shown, BF1 and BR2 used in this study are highlighted in red.
(EPS)

**S2 Fig. Overlap between morphology and DNA per sample.** The overlap between morphology and DNA (in yellow), as well as the fractions of taxa only detected with DNA (blue) and morphology (red), for each of the 25 samples separately, as well as averages (last column), separated for each of the six taxa groups.
(EPS)

**S3 Fig. Specimen and read abundances.** Relative abundances of (A) specimens in the traditional morphological assessment and reads in the metabarcoding data of (B) separately sequenced taxa groups combined and (C) samples pooled prior to amplification. In addition to the six groups assessed in this study, the fractions of water mites (in morphology), as well as vertebrates and unidentified MOTUs (in DNA data) have been included.
(EPS)

**S4 Fig. EQR scores without water mites.** Comparison of EQR scores for the morphological data with and without water mites (ACA). No DNA was obtained from water mites due to the buffer they were stored in. Pearson correlation value provided in the panel, $p < 0.001$.
(EPS)

**S1 File. Taxon lists.** Taxon lists for the three datasets: Morphologically identified taxa (with specimen counts), DNA-based identifications from the sorted samples, and DNA-based identifications form the pooled samples (both with read counts).
(XLSX)

## Acknowledgments

We thank Aquon and Wouter Balster for supplying the samples and morphological identifications, as well as Bart Schaub of Hoogheemraadschap van Rijnland for providing us with access to the collection of WFD samples.

## Author Contributions

**Conceptualization:** Kevin K. Beentjes, Arjen G. C. L. Speksnijder, Berry B. van der Hoorn.

**Data curation:** Kevin K. Beentjes, Marten Hoogeveen.

**Formal analysis:** Kevin K. Beentjes.

**Funding acquisition:** Berry B. van der Hoorn.

**Investigation:** Kevin K. Beentjes, Rob Pastoor.

**Methodology:** Kevin K. Beentjes.

**Software:** Marten Hoogeveen.

**Supervision:** Arjen G. C. L. Speksnijder, Menno Schilthuizen, Berry B. van der Hoorn.

**Visualization:** Kevin K. Beentjes.

**Writing – original draft:** Kevin K. Beentjes.

**Writing – review & editing:** Kevin K. Beentjes, Arjen G. C. L. Speksnijder, Menno Schilthuizen, Marten Hoogeveen, Rob Pastoor, Berry B. van der Hoorn.

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
