## [Decision Letter · Decision Letter 0]

29 Jul 2019

PONE-D-19-18500

Increased performance of DNA metabarcoding of macroinvertebrates by taxonomic sorting

PLOS ONE

Dear Drs. Beentjes,

Thank you for submitting your manuscript to PLOS ONE. After careful consideration, we feel that it has merit but does not fully meet PLOS ONE’s publication criteria as it currently stands. Therefore, we invite you to submit a revised version of the manuscript that addresses the points raised during the review process.

1) Both reviewers have concerns and comments on the approach to handle the differences in the sequence read depths between different samples.   

2) Reviewer #2 raises an important point in that it seems a single primer pair was used for the amplification reactions that may work very inefficiently for some of the phyla being investigated.  The authors should elobarate on the potential shortcomings of this approach, and (if feasible) evaluate the differences of using a different primer pair on a test sample.  

3) All minor concerns of both reviewers should be addressed by making appropriate changes to the text of the manuscript.  

We would appreciate receiving your revised manuscript by Sep 12 2019 11:59PM. To enhance the reproducibility of your results, we recommend that if applicable you deposit your laboratory protocols in protocols.io, where a protocol can be assigned its own identifier (DOI) such that it can be cited independently in the future. For instructions see: http://journals.plos.org/plosone/s/submission-guidelines#loc-laboratory-protocols

We look forward to receiving your revised manuscript.

Kind regards,

Sebastian D. Fugmann, Ph.D.

Academic Editor

PLOS ONE

Journal Requirements:

2. We note that you are reporting an analysis of a microarray, next-generation sequencing, or deep sequencing data set. PLOS requires that authors comply with field-specific standards for preparation, recording, and deposition of data in repositories appropriate to their field. Please upload these data to a stable, public repository (such as ArrayExpress, Gene Expression Omnibus (GEO), DNA Data Bank of Japan (DDBJ), NCBI GenBank, NCBI Sequence Read Archive, or EMBL Nucleotide Sequence Database (ENA)). In your revised cover letter, please provide the relevant accession numbers that may be used to access these data. For a full list of recommended repositories, see http://journals.plos.org/plosone/s/data-availability#loc-omics or http://journals.plos.org/plosone/s/data-availability#loc-sequencing

Reviewers' comments:

Reviewer's Responses to Questions

**Comments to the Author**

1. Is the manuscript technically sound, and do the data support the conclusions?

Reviewer #1: Partly

Reviewer #2: Partly

2. Has the statistical analysis been performed appropriately and rigorously? 

Reviewer #1: Yes

Reviewer #2: Yes

3. Have the authors made all data underlying the findings in their manuscript fully available?

Reviewer #1: Yes

Reviewer #2: Yes

4. Is the manuscript presented in an intelligible fashion and written in standard English?

Reviewer #1: Yes

Reviewer #2: Yes

5. Review Comments to the Author

Reviewer #1: The authors present a methodological analysis of DNA metabarcoding, using a set of 25 freshwater arthropod community samples, with a particular focus on taxonomic composition of samples. They present a comprehensive pipeline by which they identify their OTUs to species level, using this data to compile molecular species lists and community indices. They have particular power in this analysis as these samples have been previously sorted and morphologically identified, against which data the authors validate their species lists and diversity/composition indices. Their main methodology is the separate DNA metabarcoding of separate taxonomic groups within each sample, compared against DNA metabarcoding of all of these groups combined within each sample. Their metabarcoding results suggest that in general, morphological and molecular species lists have considerable overlap, but metabarcoding misses a substantial proportion of species. Their key conclusion is that this observation is exacerbated in pooled samples containing all taxa, even when rarefying to ensure standardisation of effective sequencing coverage, and thus that metabarcoding taxa separately is important for full community recovery.

This research is largely well presented, although I feel the depth of analysis is somewhat shallow in places. In particular, the authors do not analyse or discuss the possible effect of sequencing depth on their findings, both in terms of the extent to which they can compare between samples and the extent to which their findings are able to be generalised. While they present the total numbers of reads recovered from sequencing, they only passingly report some of the post-filtering and bioinformatics per-sample reads and ranges. Rarefaction is only passingly discussed (in the results section), and the reader's understanding of sample treatment is confounded by unclear description of the way that PCR replicates were combined. While it is clear, and largely justified, that the authors rarefied the read counts of the pooled samples to be equal to 6 times those of the separate-taxon samples, it is not stated what these rarefaction targets were, nor unequivocally stated whether all separate-taxon samples were rarefied to the same target. I am sure the authors agree that this is a crucial step if comparing lists of recovered species or summary indices. I suggest that the authors clarify how exactly technical replicates are treated as well, as it is unclear whether data from these were combined pre- or post- rarefaction. The authors' description of the 'additive' combination of PCR replicates implies that only species lists were combined; if so, this is somewhat concerning given they report several sequencing failures - I don't feel that species lists from samples with two successful replicates would necessarily be comparable to samples with only one. A more valid approach would see the data from multiple replicates being combined before each independent sample is rarefied to the same rarefaction target (or discarded for having lower than the target number of reads). Finally on this, the authors report that some of the source 25 samples did not have all six taxa: should the pooled samples from these not be rarefied to the appropriate lower multiple to make this more comparable?

More widely, beyond simply reporting per-sample read numbers and rarefaction targets, the authors could improve the extent to which their results can be generalised by examining the effect of different read numbers on their findings - assuming they have sufficient coverage in the first place. This could be achieved by utilising a range of rarefaction targets to simulate reduced sequencing coverage, and examining how the values presented in figure 2 respond. This could all be simulated based on existing data. The authors may be interested in a recent paper (doi: 10.1111/1755-0998.13008) that suggests that protocol variation can have substantial effects on community recovery if coverage does not reach an asymptote (although not specifically regarding taxon composition)

Otherwise, I have no other major comments on this nice paper, but there are various presentational and detail aspects which hinder the reader's easy understanding of the subject matter and the authors' comprehensive discussion of their findings:

Abstract:

The authors define WFD (line 20) but not EQR (line 30). "Such as" on line 22 seems misplaced. Otherwise clear and succinct.

Introduction:

The introduction is short, but this is fine, it provides a clear background and well stated aims without waffle. I would suggest that line 46: "Until this day" be changed to "To date" as a more standard English phrase.

Methods:

>Sample selection and processing:

Line 77: Citation needed for "identified according to national WFD monitoring standards" - what are these methods/standards?

Line 79: Some explanation for the EQR score range and quality classes would improve reader understanding here - at the very least a citation for the reader to find more, but an in-text explanation preferred.

It is unclear to the non-Dutch reader what exactly Aquon is - in line 76 I assumed it was a specific location, and was confused later that a location was performing identification. It took context clues from later (and a google search) to understand that it was a freshwater ecological surveying company. I suggest reorganising this paragraph to make this clearer: for example, "Freshwater arthropod samples were collected using [method] in [location] by [ecological survey company] Aquon on [dates] according to [standardised WFD monitoring guidelines]. Aquon taxonomists sorted and identified these samples into [groups]. For this study, we selected 25 samples based on ratio ... etc"

It's not crucial, but some useful context could be supplied if the authors report the total number of samples from which the 25 were selected, and more importantly the overall EQR and quality class ranges from which their sample EQR and quality class ranges were decided. It's currently unclear to what extent their samples provide a wide sample.

On line 90, the authors report 158 tubes. 25 samples of 6 sample groups used gives a total of 150 - where do the extra 8 come from?

>DNA extraction and amplification:

The authors do not report the number of species, or more crucially the numbers of individuals or the biomass ranges of species within tubes, nor the extent to which this varies between samples of the same taxon or different taxa. Considerable variation in these values may introduce biases to molecular recovery rates and diminish the validity of their findings. The authors mention the effect of biomass on OTU recovery in passing in their introduction, and again in their discussion, but do not report any attempts to mitigate these effects.

>Bioinformatics

This section could be improved by more detail to improve reproducability of the methods. For each step of the bioinformatics process, the authors should fully report the parameters used, or that default parameters were used. Furthermore, there should be some justification for software and parameter choices made. No denoising was performed, and reads shorter than the target length were permitted but not reads longer. Length variants were permitted that allowed frameshift mutant sequences (length variants not a multiple of 3 bases different from the target length) - these are likely to be sequencing errors rather than true sequences. It is not clear at what point sequences were converted from fasta to fastq? It is also unclear whether OTU clustering was performed on a by-sample basis or over the entire dataset.

Line 161: "Anticipating on" should just be "Anticipating"

>Comparison morphology versus DNA

The title of this section is not good English. It should be something along the lines of "Comparison of morphology vs molecular identification"

Line 181: what is meant by "accumulated" in the context of shared taxonomic identity. Does this just mean merged? It would be illuminating if the authors provided some analysis or commentary on what the recovery of multiple OTUs per species says about the OTU clustering methods, particularly whether this varied between taxa

Line 182: Again, what is meant by "accumulated" in this context?

There could be a little more discussion of ecological quality ratios: what are these actually measuring? Was morphological data also converted to presence-absence? Studies into methodological considerations in metabarcoding are relevant to ecologists from a wide range of backgrounds, not just those working in freshwater or marine monitoring. Some of this is discussed in more detail in the discussion, but until then a reader unfamiliar with EQRs is left in the dark.

Results

As supplied, having figure captions in the text but the figures at the end of the document is a bit of a pain to read. It would be easier to have the figures in the text, or at least the captions with the figures at the end.

The results are by large comprehensively and clearly reported. There is a substantial amount of detail here that is perhaps unnecessary, and the reader can easily lose track of the main finding, the comparison between the combined taxon-specific samples and the pooled samples in terms of recovered community composition. In particular the reportage of taxonomic composition is a bit dry and repetitive, and figure one is not particularly surprising or informative. Perhaps the authors could find a more engaging way to present taxonomic composition and move figure one to the supplement - entirely their choice though.

Figure two as supplied is portrait and difficult to read. Furthermore, it appears to be missing the lower standard error bars? The response axis should read richness rather than diversity, to correspond with the language in the text and be unequivocal.

Line 267 is missing a closing parathesis

The rate of uncertainty in taxonomic identification at both morphological and molecular level is unsurprising, and the authors should be applauded for undertaking considerable work in designing a pipeline to achieve the best possible molecular identifications. However, this does beg the question: to what extent might the findings be different if the authors focused only on species/OTUs that can be definitively identified to species level by both morphological and molecular methods. I.e. rather than

What happens if drop out comparisons of presence/absence of any uncertain taxon, and focus solely on clearly identified species at both DNA and morphological level? I.e. the dropping of any entry in category 2-4 from the lines 191-200. At very least, it would be useful if the authors could report the number of entries falling into each category to reassure the reader that uncertainty levels were low

I would suggest bringing Supplemental Figure 4 into the main text, this quite strikingly shows the contribution of the annelids to the variation in EQR scores between Fig 3B and 3C-H. The authors may wish to consider simplifying and perhaps even merging Figures 3 and 4 by removal of the coloured blocks and simplification of axes etc.

Discussion

The discussion is detailed, thoughtful and well referenced, with the conclusions well supported by the results. It could perhaps be reduced in length somewhat.

In line 364, the authors report that size-based sorting affects recovery rates, yet they do not justify why they did not do this.

The authors do not compare their findings to recent work also looking at the effects of taxonomic composition on OTU recovery in metabarcoding, which did not use morphological data as a comparison but instead used pools of increasing taxonomic complexity, finding somewhat similar results (doi: 10.1002/ece3.4839).

I congratulate the authors on an interesting paper, and am sure it will be an important contribution to metabarcoding and molecular ecology with the above comments addressed.

Reviewer #2: The authors present an interesting paper evaluating the performance of metabarcoding versus morphological assessment of aquatic invertebrate samples. Although the general premise of the paper is interesting, I have a few concerns on the methodological and technical aspects of the paper. My first concern is the use of only a single primer set (BF1/BR2) to assess a community not only containing insects, but crustaceans, annelids, and molluscs. This primer is not well suited for organisms from different phylums and I would argue that is a poor primer choice for non-insects. The poor performance of this primer pair on the pooled sample is unsurprising given the diversity of organism (or templates) present in the samples. My other major concern is that the authors try to account for uneven read depth by taking one six of the read depths to create pseudoreplicates of pooled samples from sorted samples (at least that is my understanding from the text). The simplest solution and most comparable solution would be to subsamples reads to have an equivalent read depth for all samples. This would allow a more meaningful comparison of recovery amongst all the samples. My third major concern is that text can be a bit confusing on the discrepancies between morphological and molecular identifications. I think it would be best to clearly state the errors in either data set (preferably morphology first) and then elaborate on the where mistakes are made in the other approach. I.e. I want to see some sort of assess of the quality of morphological ids and the reference library. This will make it easier to then sort out differences in recovery between the two approaches. Otherwise, I did appreciate the authors use of the top 100 hits to assign taxonomy to MOTUs. This is a great approach that will hopefully become more common in the field and is a strength of the paper. I have a few other comments below:

Line 56: could probably add a few more references for mock communities and maybe argue that there is a greater need to use actual samples because they tend to be a bit naïve. I.e. the diversity present in your samples is a testament to the limitations in diversity often present in mock samples.

Line 100 – 102: This sentence is quite awkward!

Lines 92 – 102: why was no estimate of DNA concentration done prior to pooling? Depending on extraction efficiency this could have introduced another massive bias. It would have also allowed you to assess the quality of samples following extraction because what if one community was more degraded then another?

Lines 116 -130: This cannot be changed after the fact but why so many PCR cycles? Also the use of AIC was great in addition to extraction blanks!

Line 142-144: Although I am good with using an additive approach, it could be argued that some of these are stochastic in nature and this approach can over count OTUs. I think this approach can be justified and maybe a sentence or two would be useful.

Line 146-159: Why not query and remove contaminants as the first step? And why not screen and remove them from your reference library first?

Figure 1: What exactly is the difference between DNA taxon richness versus MOTU richness. I think there is a mistake in the caption that should be reviewed. I am assuming one is supposed to be morphological richness? I would just review terminology through the material and methods as well as the results.

Figure 2: have “observed diversity observed” maybe change the second observed to present?

Line 298: So the results are mutually exclusive, which on was correct or are they both wrong?

Line 351-353: what were the quality of hits?

Lines 435-439: I would argue that this is due to poor primer choice for this group.

Conclusions: I agree that things are not always clear cut but can you make a more concrete statement. Does metabarcoding provide reliable data for EQR scores or do both approaches have fundamental flaws …

6. PLOS authors have the option to publish the peer review history of their article (what does this mean?). If published, this will include your full peer review and any attached files.

Reviewer #1: No

Reviewer #2: No

---

## [Author Response · Author response to Decision Letter 0]

12 Sep 2019

We would like to thank the editor and reviewers for considering our submission, and for their constructive feedback on the manuscript. We have incorporated some changed based on their suggestions in the revised manuscript.

One of the main issues was the lack of clarity regarding the rarefaction of data we performed to address the uneven sequencing depths between pooled and sorted samples. We have added a paragraph to the methods section detailing our approach. 

While we can understand the concerns about the primer choice by reviewer #2, we wanted to perform amplifications with a single set of primers, mostly since we are trying to work towards a standard way of processing monitoring samples that should not be more complicated and time consuming than the current morphological assessments. By our knowledge, the primers selected perform well on a wide variety of macroinvertebrates, both in theory and in practice. We have assessed several primer sets prior to this project, and these primers showed most potential across the different taxonomic groups. We have included a statement about the primers in the discussion.

Based on suggestion by reviewer #1, we have combined the data from figure S4 into figure 3. This was certainly a useful suggestion, as both figures displayed similar information. Figure S4 has been changed to only detail the comparison of EQR scores with and without water mites, as that group was not included in the DNA analysis. We have also changed figure 2, to also provide the lower error bars, and we have removed the values of the fractions listed in the figure.

A few minor other edits were made to the manuscript and the figures and their legends to address the other concerns of the reviewers, and have provided a response to all comments in the "Response to Reviewers" document.

---

## [Decision Letter · Decision Letter 1]

28 Oct 2019

PONE-D-19-18500R1

Increased performance of DNA metabarcoding of macroinvertebrates by taxonomic sorting

PLOS ONE

Dear Drs. Beentjes,

Thank you for submitting your manuscript to PLOS ONE. After careful consideration, we feel that it has merit but does not fully meet PLOS ONE’s publication criteria as it currently stands. Therefore, we invite you to submit a revised version of the manuscript that addresses the points raised during the review process.

1)  The Reviewer raises only minor points that should be addressed by making appropriate changes to the text of this manuscript.  

We would appreciate receiving your revised manuscript by Dec 12 2019 11:59PM. To enhance the reproducibility of your results, we recommend that if applicable you deposit your laboratory protocols in protocols.io, where a protocol can be assigned its own identifier (DOI) such that it can be cited independently in the future. For instructions see: http://journals.plos.org/plosone/s/submission-guidelines#loc-laboratory-protocols

We look forward to receiving your revised manuscript.

Kind regards,

Sebastian D. Fugmann, Ph.D.

Academic Editor

PLOS ONE

Reviewers' comments:

Reviewer's Responses to Questions

**Comments to the Author**

1. If the authors have adequately addressed your comments raised in a previous round of review and you feel that this manuscript is now acceptable for publication, you may indicate that here to bypass the “Comments to the Author” section, enter your conflict of interest statement in the “Confidential to Editor” section, and submit your "Accept" recommendation.

Reviewer #1: (No Response)

2. Is the manuscript technically sound, and do the data support the conclusions?

Reviewer #1: Yes

3. Has the statistical analysis been performed appropriately and rigorously? 

Reviewer #1: Yes

4. Have the authors made all data underlying the findings in their manuscript fully available?

Reviewer #1: Yes

5. Is the manuscript presented in an intelligible fashion and written in standard English?

Reviewer #1: Yes

6. Review Comments to the Author

Reviewer #1: Thank you to the authors for providing a detailed response to my comments and supplying a revision of their manuscript. I am satisfied that the vast majority of my concerns have been addressed.

I have only relatively minor comments on this revision that is otherwise suitable for publication. Note that my line numbers refer to the tracked changes version of the manuscript.

Firstly, there is inconsistency in the reporting of the fragment length. On line 127 you state the length of the fragment amplified is 316bp, in agreement with Elbrecht and Leese and supplemental figure S1. However, in the response to reviewers you state the length is 313 and that you allowed a +/- 3-base variation in length around this, giving the 310-316bp stated on line 160. Which is it? It seems as if you may have allowed sequences up to 6bp shorter than the target, but no longer. In the slippage paper cited, the forward primer was observed to reduce the length by 1-2 bases, and they did not study the reverse primer. While allowing for primer slippage by using a loose length threshold may retain true biological sequences, it may simultaneously allow PCR or sequencing errors through to create erroneous MOTUs that may inflate your MOTU richness. In my opinion this concern should at least be noted and discussed in the paper, even if the authors choose to retain this less conservative length variation threshold.

Otherwise, I suggest the authors review the paper for language once again, I noticed relatively frequent typos or slight English errors. For example on line 30 "ration" should be "ratio" and line 60 "organisms groups" is not correct English. Similar minor errors are found throughout the paper. This doesn't detract from the understanding of the paper, simply an aesthetic concern.

On line 116, the authors should report the exact speed or frequency of the homogeniser used, rather than stating the maximum was used, as different devices may have different maxima. According to this https://www.ika.com/en/Products-Lab-Eq/Dispersers-Homogenizer-csp-177/ULTRA-TURRAX-Tube-Drive-Technical-Data-cptd-3646000/ it's 6000rpm

I should note that I broadly agree with reviewer 2's comments on primer choice, something I clearly didn't think about enough on my first review. I believe the current methodology provides sufficiently supported and interesting results for publication, however there is certainly some scope for discussion of primer choice. After all, the authors advocate for separate amplification of taxon pools, negating the need for a single general multi-taxon-appropriate primer-pair (or even locus). In the response to reviewers, the authors provide good reasoning for why the primers are not to blame for poor amplification of molluscs, however there seems no reason why such a discussion cannot be also included in the text of the paper itself, to acknowledge that primer choice may have been an source of error in the study, discuss whether this is the case and propose future methodological directions to ameliorate this issue.

On the other hand, I agree with the authors that their pooling on volume instead of concentration was the appropriate route to take for accurate reflection of a multi-taxon extraction.

Otherwise, the authors have improved the clarity of the text, especially on the issue of rarefaction, and overall the paper reads well, the work is well-justified and results, discussion and conclusion well-supported.

7. PLOS authors have the option to publish the peer review history of their article (what does this mean?). If published, this will include your full peer review and any attached files.

Reviewer #1: No

---

## [Author Response · Author response to Decision Letter 1]

27 Nov 2019

Response to reviewers

We would like to thank the editor and reviewers for considering our re-submission. We have adjusted the manuscript based on the remaining points raised by the reviewer.

We ran some of the analysis again based on an inconsistency noticed by the reviewer in the length used to trim the sequence data. To be certain we went back to the data and noticed it was not a typo in the manuscript, but we did indeed use incorrect values for trimming. After comparing the new results to the earlier ones, we found only small differences in the MOTU clustering, leading to a few additional MOTUs and changes in the number of reads for a selection of MOTUs. However, we found no additional taxa, other than some MOTUs identified at genera level that were already found at species level, and rare cases vice versa. Since we used a taxonomic aggregation and removed redundant taxa (as described in the manuscript) prior to comparison with morphology, we observed no changes in the analyses beyond the creation of the MOTU table. Please note that we have adjusted the numbers in the manuscript to reflect the new MOTU table with the corrected sequence filtering, and also updated Figure 1 to show the new MOTU richness, as well as supplementary File S1.

We have provided a response to the other comments by the reviewer below.

Review Comments to the Author

Reviewer #1: Thank you to the authors for providing a detailed response to my comments and supplying a revision of their manuscript. I am satisfied that the vast majority of my concerns have been addressed. I have only relatively minor comments on this revision that is otherwise suitable for publication. Note that my line numbers refer to the tracked changes version of the manuscript.

Firstly, there is inconsistency in the reporting of the fragment length. On line 127 you state the length of the fragment amplified is 316bp, in agreement with Elbrecht and Leese and supplemental figure S1. However, in the response to reviewers you state the length is 313 and that you allowed a +/- 3-base variation in length around this, giving the 310-316bp stated on line 160. Which is it? It seems as if you may have allowed sequences up to 6bp shorter than the target, but no longer. In the slippage paper cited, the forward primer was observed to reduce the length by 1-2 bases, and they did not study the reverse primer. While allowing for primer slippage by using a loose length threshold may retain true biological sequences, it may simultaneously allow PCR or sequencing errors through to create erroneous MOTUs that may inflate your MOTU richness. In my opinion this concern should at least be noted and discussed in the paper, even if the authors choose to retain this less conservative length variation threshold.

Our response: We checked the raw data and we indeed made a mistake in the analysis rather than in the manuscript. We have ran the analysis again as discussed above.

Otherwise, I suggest the authors review the paper for language once again, I noticed relatively frequent typos or slight English errors. For example on line 30 "ration" should be "ratio" and line 60 "organisms groups" is not correct English. Similar minor errors are found throughout the paper. This doesn't detract from the understanding of the paper, simply an aesthetic concern.

Our response: Several mistakes throughout the paper were corrected and sentences were rewritten for clarity. 

On line 116, the authors should report the exact speed or frequency of the homogeniser used, rather than stating the maximum was used, as different devices may have different maxima. According to this https://www.ika.com/en/Products-Lab-Eq/Dispersers-Homogenizer-csp-177/ULTRA-TURRAX-Tube-Drive-Technical-Data-cptd-3646000/ it's 6000rpm

Our response: We have added the speed used (indeed 6000 rpm) to the methods section.

I should note that I broadly agree with reviewer 2's comments on primer choice, something I clearly didn't think about enough on my first review. I believe the current methodology provides sufficiently supported and interesting results for publication, however there is certainly some scope for discussion of primer choice. After all, the authors advocate for separate amplification of taxon pools, negating the need for a single general multi-taxon-appropriate primer-pair (or even locus). In the response to reviewers, the authors provide good reasoning for why the primers are not to blame for poor amplification of molluscs, however there seems no reason why such a discussion cannot be also included in the text of the paper itself, to acknowledge that primer choice may have been an source of error in the study, discuss whether this is the case and propose future methodological directions to ameliorate this issue.

Our response: We have added several parts of the discussion in the rebuttal letter on this topic to the main manuscript.

On the other hand, I agree with the authors that their pooling on volume instead of concentration was the appropriate route to take for accurate reflection of a multi-taxon extraction.

Otherwise, the authors have improved the clarity of the text, especially on the issue of rarefaction, and overall the paper reads well, the work is well-justified and results, discussion and conclusion well-supported.

---

## [Editor Report · Decision Letter 2]

3 Dec 2019

Increased performance of DNA metabarcoding of macroinvertebrates by taxonomic sorting

PONE-D-19-18500R2

Dear Dr. Beentjes,

We are pleased to inform you that your manuscript has been judged scientifically suitable for publication and will be formally accepted for publication once it complies with all outstanding technical requirements.

With kind regards,

Sebastian D. Fugmann, Ph.D.

Academic Editor

PLOS ONE
---

## [Editor Report · Acceptance letter]

9 Dec 2019

PONE-D-19-18500R2 

Increased performance of DNA metabarcoding of macroinvertebrates by taxonomic sorting 

Dear Dr. Beentjes:

I am pleased to inform you that your manuscript has been deemed suitable for publication in PLOS ONE. Congratulations! Your manuscript is now with our production department. 

With kind regards,

on behalf of

Dr. Sebastian D. Fugmann 

Academic Editor

PLOS ONE